# Unraveling the In Vivo Protein Corona

**DOI:** 10.3390/cells10010132

**Published:** 2021-01-12

**Authors:** Johanna Simon, Gabor Kuhn, Michael Fichter, Stephan Gehring, Katharina Landfester, Volker Mailänder

**Affiliations:** 1Max Planck Institute for Polymer Research, Ackermannweg 10, 55128 Mainz, Germany; simonjohanna01@gmail.com (J.S.); gaborkuhn@gmx.de (G.K.); landfest@mpip-mainz.mpg.de (K.L.); 2Dermatology Clinic, University Medical Center of the Johannes Gutenberg-University Mainz, Langenbeckstr 1, 55131 Mainz, Germany; 3Children’s Hospital, University Medical Center, Johannes Gutenberg University, 55128 Mainz, Germany; fichter@uni-mainz.de (M.F.); Stephan.Gehring@unimedizin-mainz.de (S.G.)

**Keywords:** protein corona, nanoparticle, in vivo, serum, plasma, biodistribution

## Abstract

Understanding the behavior of nanoparticles upon contact with a physiological environment is of urgent need in order to improve their properties for a successful therapeutic application. Most commonly, the interaction of nanoparticles with plasma proteins are studied under in vitro conditions. However, this has been shown to not reflect the complex situation after in vivo administration. Therefore, here we focused on the investigation of magnetic nanoparticles with blood proteins under in vivo conditions. Importantly, we observed a radically different proteome in vivo in comparison to the in vitro situation underlining the significance of in vivo protein corona studies. Next to this, we found that the in vivo corona profile does not significantly change over time. To mimic the in vivo situation, we established an approach, which we termed “ex vivo” as it uses whole blood freshly prepared from an animal. Overall, we present a comprehensive analysis focusing on the interaction between nanoparticles and blood proteins under in vivo conditions and how to mimic this situation with our ex vivo approach. This knowledge is needed to characterize the true biological identity of nanoparticles.

## 1. Introduction

It is now widely accepted that upon contact of nanoparticles with biological fluid (e.g., blood plasma) proteins rapidly adsorb to the surface forming the biomolecular corona [1,2]. On one side, this process highly affects the physico-chemical properties of the nanoparticles and on the other side; it eventually determines their biological behaviour [3,4,5].

Depending on the nanoparticle system, a direct link between cellular interaction and specific protein was proven. For example, poly(butyl cyanoacrylate) nanoparticles which were coated with apolipoprotein E were able to cross the blood-brain barrier [6,7]. Recently, for example, the dynamic interaction of a single protein (influenza A hemagglutinin) and exosomes was studied by high-speed high-speed atomic force microscopy [8]. Next to this, due to corona formation, the colloid stability and biocompatibility of inorganic Au@Fe_3_O_4_ Janus particles was enhanced [9]. In strong contrast to this, several reports recognized that protein adsorption can significantly influence the targeting properties of nanoparticles [10,11]. Here, it was shown that there was no cellular recognition of transferrin-functionalized nanoparticle after corona formation, meaning that the targeting ligands is completely covered up by plasma proteins [12]. Therefore, it is of urgent need to understand and control the interaction of nanoparticles upon contact with blood proteins in order to improve their therapeutic efficiency [13].

Most in vitro studies utilize serum or plasma to investigate protein adsorption on nanoparticles [14]. Depending on the specific protein source (human vs. mice or serum vs. plasma) [15] there have been significant difference noticed in terms of cellular interaction and protein corona composition [16,17]. Therefore, it is of great importance to always use the respective plasma source of the chosen organism for in vitro studies (e.g., mouse cells = mouse plasma, human cells = human plasma). Overall, the in vitro experiments give the opportunity to characterize general interaction between nanoparticles and proteins. However, it is still under investigation to what extent this can reflect in vivo situation [18]. Recently, Chen et al. reported the highly dynamic nature of the in vivo protein corona [19]. They showed that complement protein (C3) bound to the nanoparticle surface; however, upon contact with the blood stream other proteins rapidly exchanged C3 from the nanoparticle surfaces. The complex nature of blood and flow velocity are therefore additionally factors, which should be taken into account in order to evaluate the nanoparticles’ behaviour under in vivo conditions [20].

Based on this, we aimed to characterize the in vivo corona of magnetic nanoparticles after they were exposed to the blood stream of mice. Our workflow allowed the direct recovery of the nanoparticles from the blood stream without the need of plasma preparation prior to nanoparticle recovery as described in other reports [21,22,23]. Nanoparticles were extracted from the blood stream via magnetic separation after a period being in the blood circulation of the living animal of up to 2 h. A thorough proteomic analysis was carried out to identify the distinct protein corona pattern. The identified corona composition was compared to an ex vivo situation using freshly extracted blood from mice or to an in vitro system, i.e., using serum or plasma with different anticoagulants. Additionally, cellular experiments were performed to correlate the corona composition with the cellular interaction of nanoparticles directly recovered from the blood or incubated under ex vivo or in vitro conditions.

## 2. Material and Methods

Nanoparticles. Magnetite nanoparticles coated with hydroxyethyl starch (named as mgHES) were obtained by MicroMod Partikeltechnologie GmbH, Rostock, Mecklenburg-Vorpommern, Germany. Nanoparticles are labelled with an IR-Dye D750 for in vivo imaging. According to the manufacture, the magnetic core makes up to 75–80% (*w*/*w*) and the solid content is 10 mg/mL. The nanoparticles were synthesized via a core shell method according to literature [24,25].

DLS. The diameter (in nm) and size distribution (PDI) of the mgHES nanoparticles was measured via dynamic light scattering (DLS, Malvern Instruments GmbH, Herrenberg, Baden-Württemberg, Germany) at 25 °C at an of 90°. mgHES nanoparticles (10 µL, 10 mg/mL) were diluted in PBS and each measurement was performed triplicate.

Zeta Potential. mgHES nanoparticles (10 µL, 10 mg/mL) were measured in a 1 mM potassium chloride solution (1 mL) at 25 °C with a Zeta Sizer Nano Series (Malvern Instruments GmbH, Herrenberg, Baden-Württemberg, Germany). Each measurement was performed in triplicate. Mean values and the standard deviations were calculated.

Mice. C57BL/6 mice were bred and maintained in the Central Animal Facility of the Johannes Gutenberg-University Mainz under pathogen-free conditions on a standard diet according to the guidelines of the regional animal care committee.

In vivo animal studies. All animals were pre-treated with 150 µL of clodronate-liposomes (Liposoma, Amsterdam, The Netherlands) and maintained for 24 h. The next day, mgHES nanoparticles (1 mg in 500 µL PBS) were administered intravenously through the tail vein. As a control, animals were treated with 500 µL of PBS. Blood was isolated via cardiac puncture after 1 min, 10 min or 1 h and supplemented with heparin. After the animal was sacrificed, the lung, spleen, liver and kidney was isolated and imaged via IVIS^®^ imaging.

In vivo protein corona. Nanoparticles were recovered from the blood stream after 1 min, 10 min or 1 h via magnetic separation. The nanoparticle pellet was washed with PBS (3 times) to remove loosely and unbound proteins. Strongly attached proteins were desorbed from the nanoparticle surface using 2% SDS (with 62.5 mM Tris*HCl). The sample was heated up to 95 °C for 5 min. Via magnetic separation the nanoparticle pellet was separated and the supernatant was analysed.

Ex vivo protein corona. Blood was isolated from C57BL/6 mice via cardiac puncture and supplemented with heparin. mgHES nanoparticle (1 mg) were added to the blood (2 mL) and incubated for 1 min. Ex vivo protein corona coated nanoparticles were isolated and purified as described above (in vivo protein corona).

Serum and plasma preparation. Blood was isolated from C57BL/6 mice via cardiac puncture and transferred into serum and plasma preparation tubes from Sarstedt, Nümbrecht, Nordrhein-Westfalen, Germany. According to the manufacture, blood was centrifuged for 5 min at 10,000× *g* (to generate serum), 5 min at 2000× *g* (to generate heparin plasma), 10 min at 2500× *g* (to generate EDTA plasma) or 10 min at 1500× *g* (to generate citrate plasma).

In vitro protein corona. Serum and plasma (EDTA, heparin, citrate) was prepared from C57BL/6 mice blood. mgHES nanoparticle (1 mg) were incubated with (1 mL) serum or plasma 1 min or 2 h incubation at 37 °C. In vitro protein corona coated nanoparticles were isolated and purified as described above (in vivo protein corona).

Pierce Assay. To determine the protein concentration, the standard Pierce Assay was performed according to the manufactures’ instruction.

SDS PAGE. Proteins (1 µg) were mixed with NuPAGE LDS sample buffer, NuPAGE sample reducing agent and applied to NuPAGE 10% Bis Tris Protein gel (Thermo Fisher, Dreieich, Hessen, Germany). The gel was run in NuPAGE MES SDS running buffer at 100 V for 1 h. SeeBlue Plus2 Pre-Stained Standard served as a molecular marker. Protein bands were visualized using the SilverQuest™ Silver Staining kit from Thermo Fisher, Dreieich, Hessen, Germanyas recommended.

In solutions digestion. Corona proteins were digested according to former instruction [26,27]. Briefly, SDS was removed from the protein samples with Pierce detergent removal columns (Thermo Fisher, Dreieich, Hessen, Germany) and proteins were were precipitated overnight using ProteoExtract protein precipitation kit (Merck KGaA, Darmstadt, Hessen, Germany) The resulting proteins pellet was re-suspended in ammonium bicarbonate (50 mM) buffer with RapiGest SF (Waters Cooperation, Eschborn, Hessen, Germany). Proteins were reduced (dithiothreitol, 5 mM, Merck KGaA, Darmstadt, Hessen, Germany) for 45 min at 56 °C and alkylated with (idoacetoamide, 15 mM, Merck KGaA, Darmstadt, Hessen, Germany) for 60 min at room temperature. A protein: trypsin ratio of 50:1 was used and the samples were incubated 14–18 h 37 °C. Finally, 2 µL hydrochloric acid (Merck KGaA, Darmstadt, Hessen, Germany) was added to quench the reaction.

Liquid chromatography coupled to mass spectrometry (LC-MS). Peptide samples were spiked with 50 fmol/µL Hi3 Ecoli (Waters Cooperation, Eschborn, Hessen, Germany) for absolute protein quantification [28] and diluted with were diluted with 0.1% formic acid. Proteomics measurements were performed with a nanoACQUITY UPLC coupled to a Synapt G2-Si mass spectrometer. Peptides were ionized with a NanoLockSpray source in positive ion mode. The Synapt G2-Si was operated in resolution mode and data-independent acquisition (MS^E^) experiments were carried out. Data was processed with MassLynx 4.1 and proteins were identified with Progenesis QI (2.0). The murine database was downloaded from Uniprot. All processing parameters are described in detail in previous reports [29,30]. Based on the TOP3/Hi3 [31] the absolute amount of each protein in fmol was determined. Further the relative amount in % based on all identified proteins was calculated.

Cell culture. RAW264.7 (DSMZ-German Collection of Microorganisms and Cell Cultures GmbH, Braunschweig, Niedersachsen, Germany) were kept in Dulbecco’s modified eagle medium (DMEM) supplemented with 10% FBS, 100 U/mL penicillin, 100 mg/mL streptomycin and 2 mM glutamine (Thermo Fisher, Dreieich, Hessen, Germany).

Cell uptake by flow cytometry. 100,000 cells were seeded out into 24-well and incubated overnight at 37 °C. Protein corona coated nanoparticles (75 µg/mL) were added to cells for 2 h, 37 °C. For flow cytometry measurements, cells were washed with PBS, detached with 0.25% Trypsin-EDTA and centrifuged (500× *g*, 5 min). Samples were resuspended with PBS and analyzed by flow cytometry via Attune NxT Flow Cytometer (Thermo Fisher, Eschborn, Hessen, Germany).

## 3. Results and Discussion

Magnetic nanoparticles (Figure 1A) which are covered with hydroxy ethyl starch (mgHES) [32,33] were injected into C57BL/6 mice. After distinct time point (1 min–2 h), blood was isolated via cardiac puncture and transferred into a heparin containing tube to prevent blood coagulation. Nanoparticles were recovered from the blood via magnetic separation in order to identify the key proteins, which adsorbed to the nanoparticles after in vivo administration. After animals were sacrificed, the in vivo biodistribution of the nanoparticles was visualized with the fluorescence imaging (IVIS^®^ imaging). Representative images for all time points are shown in Figure 1B, indicating a strong fluorescent signal from the mgHES nanoparticle treated animals in the liver.

The remaining amount of the nanoparticles in the blood was quantified by fluorescent measurements after the different time point to investigate the blood circulation time. After 10 min ~80% of the nanoparticles were detected in blood. Even after 2 h, ~35% of the initial amount of nanoparticles remained in the blood. There are two main explanations for this relatively long blood circulation. First, before mgHES nanoparticles were injected into mice, all animals were pre-treated with clodronate-liposomes to reduce the overall amount of macrophages in vivo (Appendix A) [34,35]. In literature, this strategy has been shown to significantly increase the blood circulation time of nanoparticles [6]. We chose this step to increase the amount of nanoparticles in the blood stream for a quantitative isolation of the mgHES nanoparticles. Secondly, the hydroxyethyl starch (HES) coating of the nanoparticles can additionally prolong the blood circulation time. Various studies proofed that nanoparticles modified with HES (HESylation) [36] showed a reduced cell interaction and prolong blood circulation times comparable to PEGylated nanoparticles [31,37]. For the here presented study, it allowed us to recover a sufficient amount of nanoparticles from the blood stream for further protein corona analysis.

The protein corona of nanoparticles isolated after in vivo administration was compared to protein corona of nanoparticles, which were incubated under in vitro condition using serum or plasma. For plasma generation different anticoagulants were chosen (heparin, EDTA or citrate). Therefore, blood was freshly isolated via cardiac puncture from mice and transferred into serum and plasma preparation tubes. After centrifugation, the supernatant was collected containing only serum/plasma proteins. In our previous studies, we already saw that the anticoagulant (in this case heparin) can influence protein adsorption and cellular uptake of nanoparticles [17]. Serum and plasma was obtained from the same mice strain used for the in vivo studies to minimize difference in the blood protein composition.

Already by SDS-PAGE (Figure 2A,B), we observed a radical different protein pattern for the in vivo corona (1 min of blood circulation, Figure 2B) compared to the in vitro coronas (1 min incubation, Figure 2A). For the in vivo corona, a broad unspecific protein pattern was observed, whereas the in vitro pattern is less complex and there are only 3–5 major protein bands. In a next step, we identified all corona proteins by LC-MS. From the absolute amount of detected corona proteins, we already saw that the in vitro and in vivo corona share only a minor number of proteins and the overall amount of in vivo corona proteins is significantly higher (Figure 2C).

As mentioned above, there have been publications, which already reported a significant difference for the in vitro protein corona pattern if nanoparticles where incubated with serum compared to plasma [38]. However, a detailed proteomic investigation considering all different anticoagulants is still missing. First of all, we focused on the five most abundant corona proteins detected after in vitro incubation and compared to the in vivo situation (Figure 2D). The overall most abundant corona protein for all in vitro conditions is serum albumin. Next to this, for all in vitro incubations, the five most abundant proteins already constitute ~50% of the total amount of nanoparticle surface adsorbed proteins. This is in strong contrast to the in vivo situation, where these corona proteins constitute only ~10% of the total amount of nanoparticle surface adsorbed proteins. This again underlines the complexity of the in vivo protein corona pattern. Comparing the corona of the different in vitro incubated nanoparticles, we found that the corona composition of citrate and EDTA is highly similar. Hence, this protein corona pattern is different after nanoparticles were incubated with serum or heparin plasma. The protein corona after serum and heparin plasma incubation is additionally enriched with calcium-binding protein and complement C3. Both proteins bind and require calcium for their functionality [33,34]. Citrate and EDTA are both chelating calcium to prevent blood clotting and to generate plasma [39]. This can explain the difference observed for the corona composition depending on the chosen anticoagulant. Additionally, the protein corona after 2 h of in vitro incubation was compared to the 1 min incubation timepoint (Appendix A). Again, the corona pattern of heparin plasma and serum is comparable and differs from the corona of citrate and EDTA plasma. Next to this, there are only minor differences in the in vitro corona proteome analysed by LC-MS after 1 min compared to 2 h of incubation (Appendix A and Appendix A).

To further characterize the in vivo corona composition, the protein pattern was monitored over time (direct recovery ~1 min of blood circulation up to 1 h). For all time points, the ten most abundant in vivo corona proteins, which constitute ~50% of the total amount of nanoparticle surface adsorbed proteins (Figure 3A). Next to this, the relative amount of these proteins does not change significantly over time. Also visualized by SDS-PAGE (Appendix A), the overall protein corona pattern was similar (1–60 min). This indicates that the corona is rapidly formed and remains stable within the first hour.

As we observed a highly significant difference for the in vitro situation compared to the in vivo situation, we aimed for a strategy, which reflects the in vivo corona proteom. We developed an approach, which we termed ex vivo. Blood was freshly isolated from mice, transferred into heparin containing tubes and nanoparticles were directly incubated with this whole blood for 1 min. The ex vivo corona was purified in analogy to the in vivo or in vitro corona. The heat map reflecting the 20 most abundant corona proteins highlights that the ex vivo corona is highly comparable to the in vivo corona. In strong contrast to these are the patterns for the in vitro corona. This indicated that the ex vivo approach reflects the in vivo situation to the greatest extent (Figure 3B and Appendix A).

The ultimate test for the influence of the protein corona is the demonstration that the adsorbed proteins change the cellular interaction of nanoparticles. As shown in a number of reports, due to corona formation, the cellular interactions of nanoparticles are strongly altered [40,41]. Therefore, we wanted to study the cellular interaction of nanoparticles coated with the in vitro corona compared to the ex vivo or in vivo corona (Figure 4). Cellular association towards the mouse macrophage cell line (RAW264.7, DSMZ-German Collection of Microorganisms and Cell Cultures GmbH, Braunschweig, Niedersachsen, Germany) was analysed by flow cytometry.

Already for the four in vitro incubated nanoparticles, we saw significant difference in the cellular interaction. In line with the corona composition, cellular internalization of citrate and EDTA coated nanoparticles is comparable. Those nanoparticles displayed a significant lower cellular association compared to heparin or serum coated nanoparticles. As we identified an enrichment of complement proteins in the corona after serum or heparin incubation, this might explain the enhanced cellular interaction of these nanoparticles. Complement proteins are widely identified in the corona of various nanoparticles and were shown to enhance interactions with immune cells [19,42]. Lastly, we saw that cellular interaction of ex vivo and in vivo incubated nanoparticles is comparable. The similarity in the proteome between the ex vivo approach and in vivo situation results in a comparable cellular interaction. Therefore, our ex vivo approach is an interesting and applicable procedure to obtain meaningful protein corona results.

Lastly, the in vivo biodistribution of the mgHES nanoparticles was monitored via fluorescent imaging (IVIS^®^) over time (Figure 5). All mice were pre-treated clodronate-liposomes with 24 h before PBS (as a control) or mgHES nanoparticles were injected. As described above, blood was collected via cardiac puncture and after the animal was sacrificed, all organs (liver, spleen, lung, and kidney; Figure 1C) were isolated and imaged.

For the individual organs, we observed the strongest signal intensity of the mgHES nanoparticles after 1 min in the liver (Figure 5A). Most interestingly, the signal decreased after 2 h. In contrast, there was a low signal intensity detected in the spleen after 1 min (Figure 5B). However, after 2 h, the signal intensity increased. This indicates the dynamic distribution of the mgHES nanoparticles over time. For the other organs (lung and liver), there was no significant change of the signal intensity over time (Figure 5C,D). In addition, the dynamic interactions of the nanoparticles within the body, represented by the time dependent change in fluorescence intensity measured for the different organs, have no influence on protein corona formation and composition as shown in Figure 3 and Appendix A.

## 4. Conclusions

In conclusion, in this study, we isolated nanoparticles magnetically directly from the blood stream after different time points. With this method, we were able to recover nanoparticles surrounded by the in vivo protein corona without extensive purification steps, which could alter the protein corona profile. We saw that the in vivo protein corona highly differed compared to in vitro incubated nanoparticles, taking into account the dynamic biodistribution process and interaction with different biological structure and organs. Additionally, we noticed that the chosen anticoagulant should be carefully considered, as we observed significant differences in the corona composition after incubation with serum or plasma supplemented with heparin, EDTA, and citrate. For our here presented ex vivo approach, we demonstrated that the corona composition was comparable to the in vivo situation. Therefore, with this study we aimed to improve the understanding of in vivo corona formation, which is urgently needed in order to reveal the nanoparticles behaviour under physiological conditions.

## Figures and Tables

**Figure 1 cells-10-00132-f001:**
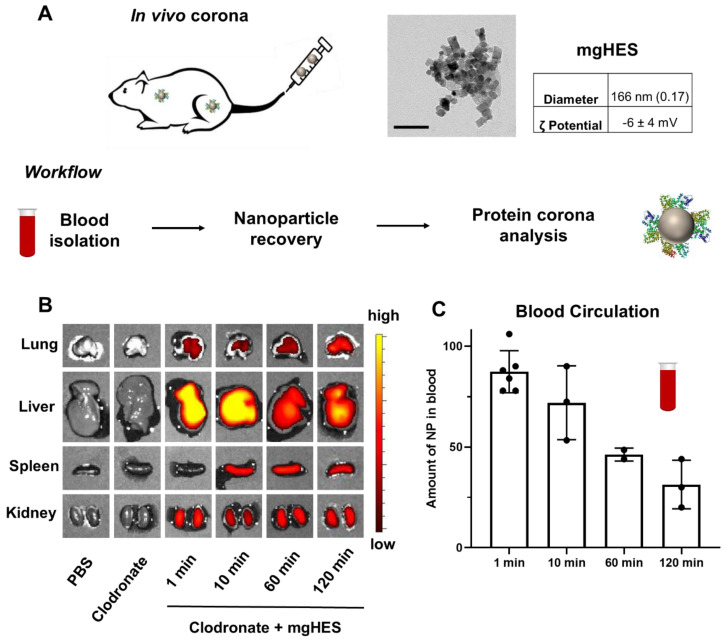
Monitoring the biodistribution and blood circulation of magnetic HES nanoparticles in vivo. (**A**) Magnetic HES nanoparticles (mgHES) were injected into mice and the blood was isolated after distinct time points (1 min–2 h). Nanoparticles were recovered from the blood via magnetic separation and further the protein corona surrounding nanoparticles after in vivo circulation was analysed. (**B**) A representative IVIS image of the different organs after treatment with PBS and clodronate-liposomes as control or mgHES nanoparticles. (**C**) The nanoparticle concentration in blood after distinct time points was measured via a Tecan Plate Reader and normalized based on the initial injected amount of nanoparticles (*n* = 2–6).

**Figure 2 cells-10-00132-f002:**
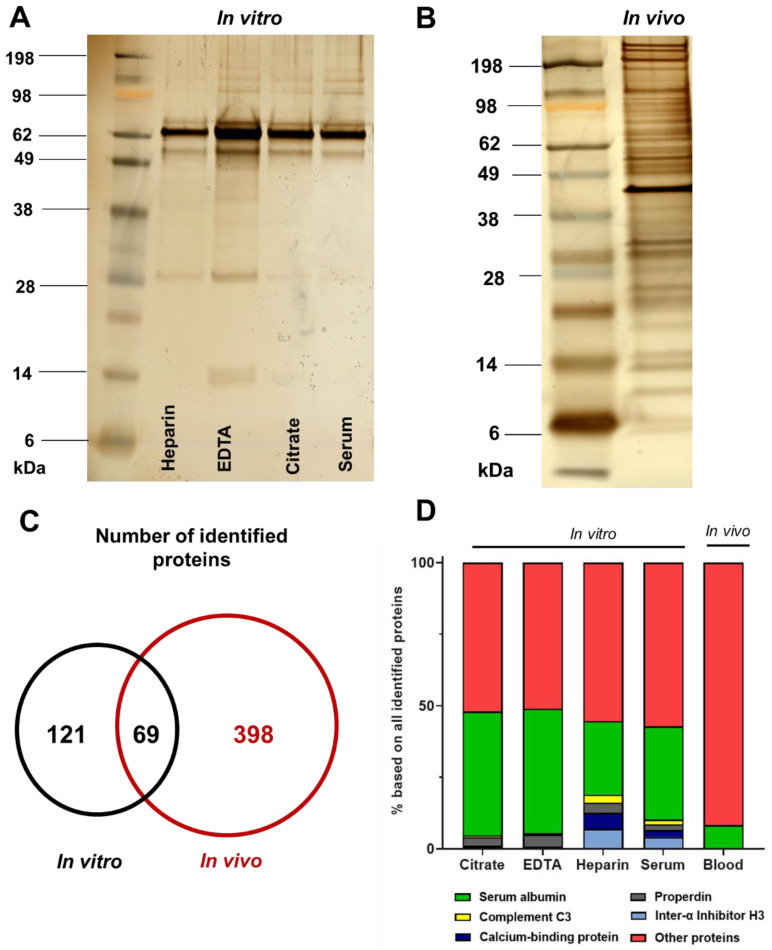
The in vivo corona pattern is not comparable with the in vitro situation. (**A**) mgHES nanoparticles were incubated with serum or plasma (heparin, EDTA or citrate) for in vitro protein corona analysis for 1 min. (**B**) Nanoparticles were recovered from the blood stream after 1 min of circulation and purified for protein corona analysis. As visualized by SDS PAGE (**A**,**B**) the pattern for the in vivo and in vitro situation highly differs. (**C**) LC-MS analysis indicates that the total number of proteins identified for the in vivo corona is significantly higher compared to the in vitro situation. Both corona types share a minor number of proteins. (**D**) The five most abundant proteins for the in vitro coronas after 1 min of incubation and their amount in the in vivo corona after 1 min of blood circulation. The average amount of each protein in % is shown and calculated from technical triplicates.

**Figure 3 cells-10-00132-f003:**
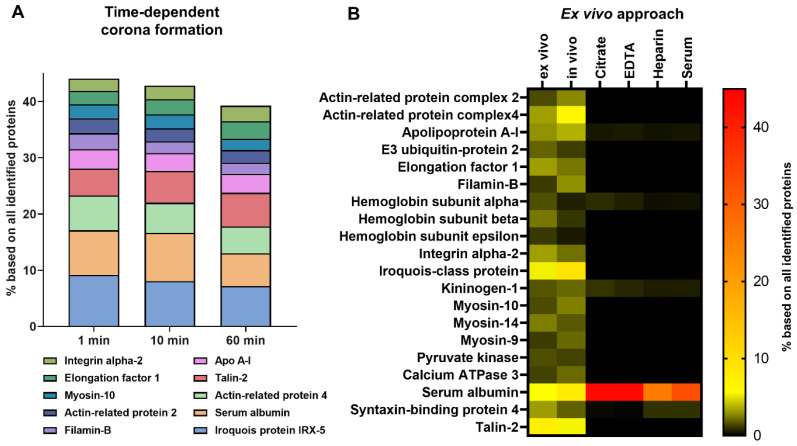
In vivo protein corona formation occurs rapidly and does not change significantly over time. (**A**) The protein corona composition in vivo was compared for three different time points (1 min, 10 min or 1 h). The average amount of each protein in % is shown (*n* = 3–5). The 10 most abundant proteins contribute to ~50 % and indicate no significant difference in their relative abundance over time. (**B**) To mimic in vivo corona formation, nanoparticles were incubated directly in blood (ex vivo) for 1 min. The heat map of the 20 most abundant corona proteins highlights the similarity between the ex vivo and in vivo corona after 1 min of blood circulation. The average amount of each protein in % is shown and calculated from technical triplicates and biological replicates for ex vivo (*n* = 2) and in vivo (*n* = 5). For comparison, nanoparticles were incubated with serum or plasma for 1 min. All identified proteins are summarized in a separate Excel sheet.

**Figure 4 cells-10-00132-f004:**
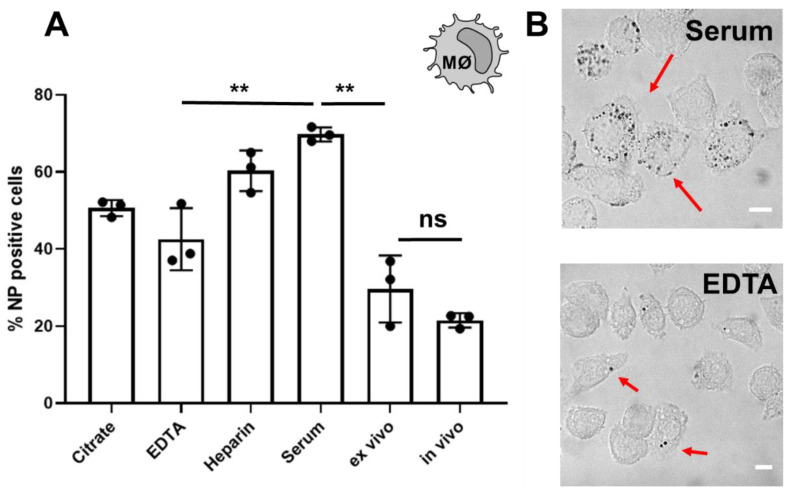
Cellular interaction of protein corona coated nanoparticles depends on the anticoagulant and differs for nanoparticles recovered from the blood stream (in vivo) (**A**) Nanoparticles were incubated with serum, plasma, blood or isolated after in vivo administration. Protein corona coated nanoparticles were subsequently added to macrophages (RAW264.7) for 2 h at a concentration of 75 µg/mL. Cellular association was analysed via flow cytometry. The amount of nanoparticle (NP) positive cells in % is shown. (**B**) Representative images illustrate the intracellular distribution of the nanoparticles. RAW264.7 cells were treated with serum or citrate plasma coated nanoparticles for 2 h at a concentration of 75 µg/mL. Scale bar: 10 µm. For statistical analysis, a student’s *t*-test (two-tailed, unpaired) was performed comparing EDTA versus serum, serum versus ex vivo and ex vivo versus in vivo incubated mgHES nanoparticle. ns = not significant, ** *p* < 0.01 (*n* = 3).

**Figure 5 cells-10-00132-f005:**
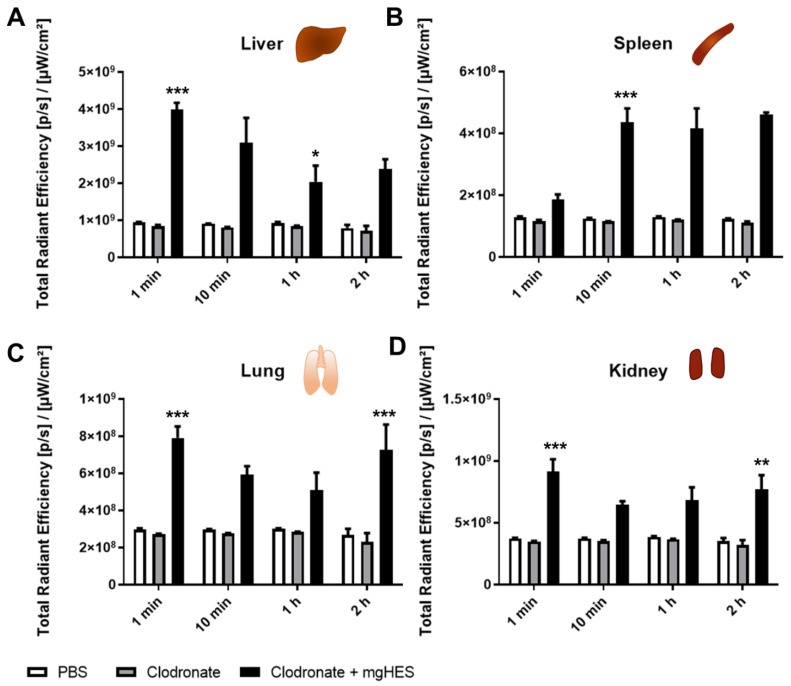
The biodistribution of the mgHES nanoparticles into different organs over time. (**A**–**D**) Animals were treated with PBS, clodronate-liposomes or mgHES nanoparticles (1 mg). Organs were isolated and imaged with IVIS^®^. The fluorescent intensity of all organs was analysed over time 1 min–2 h. (**A**) = Liver, (**B**) = Spleen, (**C**) = Lung, (**D**) = Kidney. For statistical analysis, a two-way ANOVA test was performed comparing PBS versus mgHES nanoparticle treated animals * *p* < 0.05, ** *p* < 0.01, *** *p* < 0.001 *(n =* 3–5).

## Data Availability

The data presented in this study are available in Appendix A.

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
