# Peer review of "Unraveling the In Vivo Protein Corona"

_cells, 2021, doi:10.3390/cells10010132_

Round 1
Reviewer 1 Report
This is a well-conducted study aimed to elucidate the composition of the corona proteins. I believe it will be of interest to a broad audience.
"The work by Simon and colleagues addresses the adsorption of proteins to nanoparticles. To date, this interaction is not fully understood although strongly influences the in vivo behaviour of the nanomaterial and affects its therapeutic activity. In this work, the corona composition of magnetic nanocarriers was analyzed in the blood of mice after intravenous administration of the nanocarriers. The data were compared to those obtained by mean of ex vivo and in vitro analyses using serum or plasma with different anticoagulants. Interesting findings of this work include the observation that the in vivo corona composition was similar to the composition resulting of the ex vivo approach and that the anticoagulant employed for in vitro assays strongly influence protein adsorption. These data are a stark reminder that researchers in the nanomedicine field need take into account the experimental conditions to study corona formation."
Author Response
We greatly appreciate the reviewer´s feedback on our presented work.
Reviewer 2 Report
This manuscript suggests a simple methodology for investigating the interaction between blood proteins and nanocarriers. The information are essential for nanocarrier design, in which subsequently improves the bioavailability of nanocarriers.
Below are my questions and comments:
The exposure of mgHES to serum/plasma in in vitro setting is completely different than that in in vivo mgHES in blood during in vivo sampling should be more than 1 minutes. Furthermore, authors did not mention whether blood collected via cardiac puncture was mixed with any anticoagulants. If not, will ongoing coagulation reaction happened after cardiac puncture affect the protein corona profile in in vivo samples?
I believe the discrepancy in protein corona profile between in vitro and ex vivo/in vivo reported here is simply due to the discrepancy in concentrations of blood protein interacted with mgHES in in vivo or ex vivo/ in vivo. Both serum and plasma (with anticoagulants) can be considered as concentrated blood proteins because they are obtained from differential centrifugation. Therefore, for in vitro samples, mgHES are exposed to very high concentrations of blood proteins. In contrast, for both ex vivo and in vivo samples, mgHES are exposed to relatively “diluted” blood proteins. As a result, serum/plasma would provide a better resolution to reveal potential proteins that interact with mgHES. The interaction of proteins other than serum albumin, such as C3 complement and calcium-binding proteins, with mgHES could occur in both ex vivo and in vivo setting too but the levels are below detection threshold. I would like to suggest the authors to profile the protein corona on mgHES isolated from serum/plasma after administration of mgHES to mice at given time points.
I would like to suggest the authors would consider other tools to observe protein interaction with nanocarriers. For example, high-speed atomic force microscopy (HS-AFM), which allow us to observe the interaction in real-time fashion at nanoscopic level. Please refer to this paper: PMID: 32787163.
-Line 69: "Representative images for all time points" , should be in Figure 1B.
-Line 100-106: please write your manuscript properly, in vivo vs in vitro comparison by WB is Figure 2A&B. The comparison made by LC-MS is Figure 2C.
-It will be nice if the authors provide a table to show the p-values for all proteins measured in Figure S3.
-Transthyretin increased after 2 hours of incubation. What is the role of it?
-Figure 4: Please change "Figure 1" to Figure 4.
-It would be nice if the authors provide similar heatmap format for both ex-vivo and in-vivo corona (1min and 2 hours). So that, readers can easily see the different corona profile among in vitro, in vivo and ex vivo
-The content in Figure 5 is similar to Figure 1 please consider to merge. Furthermore, the authors should not compare betweeen PBS/Clodronate liposome/mgHES because only mgHES is labeled with fluorescent dye. PBS/Clodronate liposome can be used as control, authors then should compare the fluorescent intensity in different organs to show the preference of the systemic distribution of mgHES.
-The statistic analysis used in Figure 4 shouldn’t be one-way-ANOVA because the authors are comparing 2 independent groups. Either Student-T test or Mann-Whitney U test should be performed.
-Please amend the figure legends in supplementary data: should be Figure S1, Figure S2 etc.
Reviewer 3 Report
This is an interesting work on who the different methodological approaches would result in different surface properties of nanoparticles on exposure plasma proteins. The findings of this work would be of great interest to diverse researchers focusing on designing, characterization and evaluation of nanoparticles. I suggest to address the following points before publication.
- In order to make sense of any of the results included in this manuscript, the authors have to provide proof of stability of the nanoparticles in the different tested biological media. Aggregation of the nanoparticles would greatly affect protein corona formation and would affect their behaviour.
- The authors mentioned in their introduction that different protein corona results and NP behaviour were previously observed earlier on using plasma from different sources. If this is the case, how your study translates to what happen in human scenarios - and how plasma composition is different on exposure to human plasma versus mice plasma. All the limitations of this study should be clearly demonstrated as well.
- Missing in the discussion is how the clodronate liposome treatment and the
surface coating of magnetic nanoparticles affect the results - The researchers used serum and plasma from the same mice strain to minimize differences. How about differences in the blood protein composition and concentration from one animal to another? This needs to be discussed.
- Figure 2D is confusing - I suggest a different to present the results.
- In many locations, I needed to make smart guesses to understand what the authors want to convey. Manuscript review by a scientific writing service would greatly benefit the manuscript.
-
"For all time points, the ten most abundant 139 in vivo corona proteins have a total amount of ~50% based on all identified proteins (Figure 3A).": Do you mean "they constitute 50% of the total mount of nanoparticle surface adsorbed proteins"?
- It is not clear for me how the conditions of the experiment qualify it to be termed ex vivo. Isn't this in vitro experiment using blood freshly withdrawn and treated with heparin?
- Uptake cannot be determined using flow cytometry - all what you get here is association or interaction with cells but you will not be able to distinguish what is inside and what's outside. Please correct this in the text and in Figure 4. (also Figure 4 was mistakenly numbered as Figure 1 in the figure title). %positive cells is not a good clear title of a y-axis for Figure 4A.
- I am not sure I understand the rationale behind determining the biodistribution of nanoparticles in the different organs in regards to the scope of this study. Also the "dynamic distribution of the nanocarriers over time" is not clear and should be further explained and the implications of this should be clearly stated.
- I don't find an excel sheet attached with the individual proteins.
Other minor comments:
- "From the absolute amount of detected corona proteins, we already saw that the in vitro .....and the overall amount of in vivo corona proteins is significantly higher (Figure 2B)." : This is a wrong citation - Figure 2B does not include LC-MS results.
- I would not call the magnetic nanoparticles as "nanocarriers", since carriers are usually used to describe a drug delivery system, so this could be confusing to the readers. I would preferably describe them as "nanoparticles"
- Several typos (e.g. AcknowledgementsL), missing verbs, unclear sentences.
Round 2
Reviewer 2 Report
The revised manuscript has addressed my concerns. I recommend its publication.
Reviewer 3 Report
authors responded to the suggestions